# A Wheeler–DeWitt Quantum Approach to the Branch-Cut Gravitation with Ordering Parameters

Benno August Ludwig Bodmann [1], César Augusto Zen Vasconcellos [2,3,*], Peter Otto Hess Bechstedt [4,5], José Antonio de Freitas Pacheco [6], Dimiter Hadjimichef [2], Moisés Razeira [7] and Gervásio Annes Degrazia [1]

[1]  Departamento de Física, Universidade Federal de Santa Maria (UFSM), Santa Maria 97105-900, Brazil; benno.bodmann@gmx.de (B.A.L.B.); gervasiodegrazia@gmail.com (G.A.D.)
[2]  Instituto de Física, Universidade Federal do Rio Grande do Sul (UFRGS), Porto Alegre 90010-150, Brazil; dimiter@ufrgs.br
[3]  International Center for Relativistic Astrophysics Network (ICRANet), 65122 Pescara, Italy
[4]  Instituto de Ciencias Nucleares, Universidad Nacional Autónoma de Mexico (UNAM), Mexico City 04510, Mexico; hess@nucleares.unam.mx
[5]  Frankfurt Institute for Advanced Studies (FIAS), 60438 Hessen, Germany
[6]  Observatoire de la Côte d'Azur, 06300 Nice, France; jose.pacheco@oca.eu
[7]  Laboratório de Geociências Espaciais e Astrofísica (LaGEA), Universidade Federal do Pampa (UNIPAMPA), Caçapava do Sul 96570-000, Brazil; moisesrazeira@unipampa.edu.br
*  Correspondence: cesarzen@cesarzen.com; Tel.: +55-(51)-98357-1902

**Abstract:** In this contribution to the Festschrift for Prof. Remo Ruffini, we investigate a formulation of quantum gravity using the Hořava–Lifshitz theory of gravity, which is General Relativity augmented by counter-terms to render the theory regularized. We are then led to the Wheeler–DeWitt (WDW) equation combined with the classical concepts of the branch-cut gravitation, which contemplates as a new scenario for the origin of the Universe, a smooth transition region between the contraction and expansion phases. Through the introduction of an energy-dependent effective potential, which describes the space-time curvature associated with the embedding geometry and its coupling with the cosmological constant and matter fields, solutions of the WDW equation for the wave function of the Universe are obtained. The Lagrangian density is quantized through the standard procedure of raising the Hamiltonian, the helix-like complex scale factor of branched gravitation as well as the corresponding conjugate momentum to the category of quantum operators. Ambiguities in the ordering of the quantum operators are overcome with the introduction of a set of ordering factors $\alpha$, whose values are restricted, to make contact with similar approaches, to the integers $\alpha = [0, 1, 2]$, allowing this way a broader class of solutions for the wave function of the Universe. In addition to a branched universe filled with underlying background vacuum energy, primordial matter and radiation, in order to connect with standard model calculations, we additionally supplement this formulation with baryon matter, dark matter and quintessence contributions. Finally, the boundary conditions for the wave function of the Universe are imposed by assuming the Bekenstein criterion. Our results indicate the consistency of a topological quantum leap, or alternatively a quantum tunneling, for the transition region of the early Universe in contrast to the classic branched cosmology view of a smooth transition.

**Keywords:** branch-cut cosmology; Wheeler–DeWitt equation; quantum gravity

## 1. Introduction

Motivated by the success of quantum mechanics (QM) and pseudo-complex general relativity (pc-GR) in incorporating the mathematics principles of existential closure and completeness [1] by extending their domains of realization, QM to the complex variables sector [2], and pc-GR to the pseudo-complex domain [3–5], branch-cut gravitation theory (BCGT), in its classical version, represents an analytically continued extension of general

relativity [6] to the complex plane [7–13]. The descriptive augmented domain of quantum mechanics by incorporating complex variables has broadened our perception of the infinitesimally small scales, with direct physical manifestations [14,15]. In turn, these notions led pc-GR, embedded in a pseudo-complex domain, to a suppression mechanism of the primordial gravitational singularity and to the prediction of existence of dark energy outside and inside cosmic mass distributions [3–5].

BCGT describes a hypothetical set of independent multiple universes existing in parallel, based on the multiverse[1] conception by Hawking and Hertog [16], each emerging from its own singularity. Imposing that the multiverses compose a single universe, in the Riemann limit, the multiple singularities merge, generating topological and complex smooth structures of foliation leafs, continuously connected, described by Riemann surfaces. The corresponding solutions of the analitically continued Einstein equations, represented by the helix-shaped branch-cut function $\ln^{-1}[\beta(t)]$, give rise to an alternative formulation of the Friedmann equations, as a function of complex time $t$, given by[2]

$$\left( \frac{\frac{d}{dt} \ln^{-1}[\beta(t)]}{\ln^{-1}[\beta(t)]} \right)^2 = \frac{8\pi G}{3} \rho(t) - \frac{kc^2}{\ln^{-1}[\beta(t)]} , \tag{1}$$

and

$$\left( \frac{\frac{d^2}{dt^2} \ln^{-1}[\beta(t)]}{\ln^{-1}[\beta(t)]} \right) = -\frac{4\pi G}{3} \left( \rho(t) + \frac{3}{c^2} p(t) \right). \tag{2}$$

Equations (1) and (2), and their corresponding complex conjugate versions, describe a smooth universe with a fine-tune transition region from contraction to expansion — purely geometric in nature, that replaces the cosmological singularity (Figure 1). Similar procedures allow to obtain analytically continued expressions for the energy-stress conservation law, Hubble rate, deceleration parameter, Ricci scalar and the Ricci curvature, as well as the corresponding complex conjugated expressions.

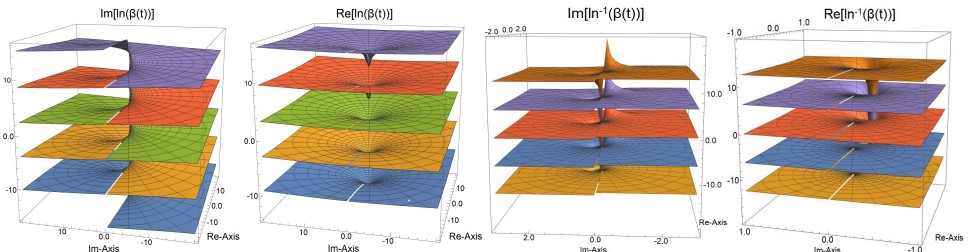

**Figure 1.** On the two left figures, characteristic plots of the Riemann surface associated with the imaginary and the real parts of the function $\ln[\beta(t)]$, the scaling in time of the branch-cut universe (the reciprocal of $\ln^{-1}[\beta(t)]$). The plot of the imaginary part shows connected glued domains: the various branches of the function are glued along the copies of each upper half plane with their copies on the corresponding lower half plane in a suitable way to make $\ln[\beta(t)]$ continuous. Each time the variable $\beta$ moves around the origin, $\ln[\beta(t)]$ moves to a different branch, with its values, on each foliaition leaf, differing from its principal value by a multiple of $2\pi i$. A similar analysis apply to $\ln^{-1}[\beta(t)]$. On the two right figures, characteristic plots of $\ln^{-1}[\beta(t)]$.

In branching gravitation, the primordial singularity is replaced by a family of Riemann foliation leafs in which the branch-cut cosmic scale factor[3] $\ln^{-1}[\beta(t)]$ shrinks to a finite critical size, shaped by the range-, foliation leafs regularization- and domain extension-$\beta(t)$-function, with its range domain above the Planck length according to the Bekenstein criterion[4] [11]. In the contraction phase, as the patch size decreases with a linear dependence on $\ln[\beta(t)]$, light travels through geodesics on each Riemann foliation leaf, circumventing continuously the branch-cut, and although the horizon size scale with $\ln^\epsilon[\beta(t)] / \ln[\beta(t)]$, where $\epsilon$ denotes the dimensionless thermodynamics connection, the length of the path to be traveled by light compensates for the scaling difference between the patch and horizon

sizes. Under these conditions, causality between the horizon size and the patch size may be achieved through the accumulation of branches in the transition region between the present state of the universe and the past events [11]. In addition to causality, the flatness and the horizon dilemmas of cosmology stand out. The flatness problem concerns the value of the ratio between the total density of the universe and the critical density resulting in a very small Planck value of the time-dependent and dimensionless cosmic spatial factor [17–19], $\Omega_c$, which scales as $\ln^{2\epsilon}[\beta(t)] / \ln^2[\beta(t)]$. The horizon problem in turn arises exactly because the patch corresponding to the observable universe was never causally connected in the past [17–19]. The restoration of causality in BCGT brings an additional perspective with a view to the future elucidation of these 'cosmological puzzles' [12].

In short, in the BCGT formalism of spacetime, a 3 + 1 dimensional Riemannian manifold $\mathcal{M}$ with metric $g$, is foliated into a one parameter family of space-like slices (leafs) or continuous trajectories (see Figure 1), with the spatial slices assumed to be closed. As a corolary, the branched gravitation approach only expands the domain of realization of the governing principles of general relativity, as well as the operations that underlie its theoretical foundations.

Recently, we have proposed a topological canonical quantum approach [13] for the classical branch-cut cosmology on basis of the renormalizable Hořava–Lifshitz theory of gravity (HLGT) [20] and the Wheeler–DeWitt Equation (WdW) [21]. HLGT is General Relativity augmented by counter-terms to render the theory regularized[5]. General Relativity is not renormalizable and therefore not applicable for very small distances, such as those associated with the beginning of the universe, central point of study at the BCGT. On the other hand, HLGT, due to its anisotropic space-time scaling, is not Lorentz invariant in the high energy UV regime. However, for small distances, the incorporation of higher order derived terms in the spatial components of the curvature to the usual Einstein–Hilbert action, gives rise to a theory free of ghosts and, therefore, HLGT is more appropriate to describe quantum effects of the gravitational field, as for instance vacuum decay processes in the early stages of the universe [24]. The parameters of the theory are the critical exponent $z$ and the foliation parameter $\lambda$, associated with a restricted foliation compatible with the Lifshitz scaling. In the low energy limit $z \to 1$ the Lorentz invariance is recovered. In the infrared limit, to recover the full diffeomorphisms symmetry and the usual foliation of the ADM formalism, the $z \to 1$ limit must be accompanied by the limit $\lambda \to 1$ [24].

The WdW equation solutions, represented by a geometric functional of compact manifolds and matter fields, describe the evolution of the quantum wave function of the Universe [25,26]. A puzzling aspect of the WdW equation however is the absence of the time variable. According to [22], the main problem with this issue in quantum gravity is perhaps its closeness to a classical space–time picture. For Rovelli [27] the absence of time is a feature of the classical Hamilton–Jacobi formulation of general relativity, and the wave function is only a function of the "3-geometry", namely the equivalence class of metrics under a diffeomorphism, and not of the specific coordinate dependent form of the metric tensor. According to the second law of thermodynamics, forward in time represents the direction in which entropy increases and in which we obtain information, so the flow of time would represent a subjective feature of the universe, not an objective part of physical reality [27]. In this realm, in which the observable universe does not exhibit time-reversal symmetry, events, rather than particles or fields, are the basic constituents of the universe, implying that the evolution of physical quantities is related to the description of the relationship between events [27–30]. For instance, given the wave function of the universe as a functional constrained to a region configuration of a super-space that contains a three-surface and matter fields, represented by $\Phi$, where the metric is described by $h_{ij}$ the corresponding WdW wave function $\Psi(h_{ij}, \Phi)$ may be interpreted as describing the evolution of $\Psi(\Phi)$ in the physical variable $\Phi$.

In this contribution we go beyond the previous formulation. The momentum operators are deduced and the quantum version of the Hamiltonian is obtained by addressing the well-known ambiguity on the ordering of operators in the Wheeler–DeWitt Hamiltonian[6].

Although there exists infinite possibilities, a parameter $\alpha$ which defines the ordering of the operators was restricted, for comparison purposes, to a special class of values, following the options of some authors. More precisely, $\alpha = 1$ [32], and $\alpha = 0, 2$ [33].

Determining the composition of matter and energy in the Universe represents one of the most important challenges in cosmology. The most recent developments suggest that the Universe's content, other than dark matter, is unaccounted for or missing. In this contribution, in addition to a branched universe filled with underlying background vacuum energy, primordial matter and radiation, in order to make contact with standard model calculations, we supplement this formulation with baryon matter, dark matter and quintessence contributions. Quintessence—a time-varying, spatially inhomogeneous, and negative pressure component of the cosmic fluid—is a dynamic ingredient: its energy density and pressure vary with time and is spatially inhomogeneous [34,35]. The main motivation to consider the presence of quintessence is to address, in the future, the so-called "coincidence problem", related to the initial conditions necessary to produce the quasi-coincidence of the densities of matter and quintessence in the present stage of the universe [34,35]. Furthermore, in the approaches commonly presented in the literature, the material composition of the primordial universe refers to the plasma of quarks and gluons and leptons, and with regard to dark matter, a frequent approach is that of a geometric effect through a cosmological constant. In this work, aiming at the future study of the matter–antimatter asymmetry of the universe and baryogenesis, as well as the dark matter described by a kind of cosmic fluid, with an equation of state of the form $P = \omega \rho$, we consider the contribution of the additional terms. The theory coupling parameters, $g_i (i = 0, 1, ..., 9)$, are in turn dimensionless running couplings constants.

Finally, the boundary conditions for the wave function of the Universe are imposed by assuming the Bekenstein criterion, which indicates the existence of an universal upper bound of magnitude $2\pi R/\hbar c$ to the entropy-to-energy ratio $S/E$ of an arbitrary system of effective radius $R$.

We proceed as follows:

- The line element squared within the branched cosmology is defined and can be retrieved in Refs. [9,13].
- The action is defined, using the Horava–Lifshitz theory of gravity, which is the General Relativity augmented by counter-terms to render the theory regularized. For more information, please consult Ref. [24]. The basic ingredients are now expressed in terms of $\ln[\beta(t)]$, which substitutes the standard scale factor $a(t)$. In Section 2.1, the classical impulse variable is defined and the classical Hamiltonian constructed.
- A quantization procedure is applied, elevating the momentum operator and Hamiltonian to operators. As a result we obtain the Wheeler–DeWitt equation.
- Following this path, a parameter $\alpha$ appears which defines the ordering of the operators, as applied in the past to the Wheeler–DeWitt equation. This leaves us with three possible equations.
- These equations are solved using the Range–Kutta numerical analysis iterative method. Unlike the approaches usually found in the literature, in our calculations we do not use approximations. We then obtain new analytic solutions, depending on the boundary conditions based on the Bekenstein's theorem, which provides an upper limit for the entropy. For more information, please consult [10–13,36].

## 2. Extended Class of the Branched Quantum Cosmological Solutions

In what follows we investigate a branched quantum formulation of the WDW equation, whose the only dynamical variable, the helix-like scale factor analytically continued to the complex plane, as well as its corresponding conjugate momentum, are raised to the rank of quantum operators.

The equation developed by Wheeler and DeWitt, in 1967, represents a fundamental approach for describing quantum gravity [21]. As stressed before, this model, based on the Arnowitt-Deser-Misner decomposition of canonical general relativity in 3 + 1 dimensions,

is additionally complemented by a boundary term proposed by the authors of Refs. [37–40]. Dirac's canonical quantization procedure applied to the Einstein–Hilbert action results in a second-order functional differential equation defined in a configuration superspace, whose solutions depend in general on a three-dimensional induced metric and matter fields [21,25,26,40].

### 2.1. Branch-Cut Formulation of the Weeler-DeWitt Equation

The complex scale factor $\ln^{-1}[\beta(t)]$ represents, in branched cosmology, as stressed before, the only dynamical variable[7]. The branched manifold $\mathcal{M}$ is in turn layered on hypersurfaces, $\Sigma_t$, which are restricted to Riemann foliation leafs, characterized by a complex time parameter, $t$, with the normalized branching line element analytically continued in 4 dimensions defined as [8,9]

$$ds^2_{[ac]} = -\sigma^2 N^2(t)c^2 dt^2 + \sigma^2 \left(\ln^{-1}[\beta(t)]\right)^2 \left[\frac{dr^2}{(1-kr^2(t))} + r^2(t)\left(d\theta^2 + \sin^2\theta d\phi^2\right)\right]. \quad (3)$$

In expression (3), the variables $r$ and $t$ represent, respectively, real and complex spacetime parameters and $k$ the spatial curvature of the multiverse, more specifically, negatively curved (k = −1), flat (k = 0) or positively curved (k = 1) spatial hypersurfaces. $N(t)$ in turn represents the lapse[8] function with $\sigma^2 = 2/3\pi$ denoting a normalisation factor.

In what follows, we consider as a starting point the renormalizable Hořava–Lifshitz theory of gravity whose action, given by $\mathcal{S}_{HL}$, employs terms dependent on the scalar curvature of the Universe and its derivatives, in different orders, defined in the form [20,41]:

$$\begin{aligned} \mathcal{S}_{HL} = \quad & \frac{M_P}{2} \int d^3x dt N \sqrt{-g} \left\{ K_{ij}K^{ij} - \lambda K^2 - g_0 M_p^2 - g_1 R - g_2 M_P^{-2} R^2 - g_3 M^{-2} R_{ij} R^{ij} \right. \\ & - g_4 M_P^{-4} R^3 - g_5 M^{-4} R(R_j^i R_i^j) - g_6 M^{-4} R_j^i R_k^j R_i^k - g_7 M_P^{-4} R\nabla^2 R \\ & \left. - g_8 M_P^{-4} \nabla_i R_{jk} \nabla^i R^{jk} \right\}; \end{aligned} \quad (4)$$

in this expression as previously informed $g_i$ denotes the running coupling constants associated to the curvature-dependent terms and its derivatives, $M_P$ represents the Planck mass, and $\nabla_i$ are the covariant derivatives. The branching Ricci components of the three dimensional metrics in Equation (4) are determined by imposing a maximum symmetric surface foliation [13]. We then obtain

$$R_{ij} = \frac{2}{\sigma^2 \ln^{-2}[\beta(t)]} g_{ij}, \quad \text{and} \quad R = \frac{6}{\sigma^2 \ln^{-2}[\beta(t)]}, \quad (5)$$

where $R$ represents the branching scalar curvature. The trace of the extrinsic curvature tensor, $K_{ij}$, which measures geometry modifications as well as the deformation rates of the normal to a hypersurface as it is transported from one point to another, corresponds to a sub-manifold, which depends on the particular embedding and takes the form (for the details see [13])

$$K = K^{ij} g_{ij} = -\frac{3}{2\sigma N} \frac{\left(\frac{d}{dt}\ln^{-1}[\beta(t)]\right)}{\ln^{-1}[\beta(t)]}. \quad (6)$$

Through the use of standard canonical procedures of quantum field theory, a Lagrangian density and the Hamiltonian of the model can be obtained (see [13,20,24,33,41–44]).

### 3. Spacetime Topological Canonical Quantization

The Lagrangian density of the model is quantized, trough a spacetime topological canonical quantisation[9], by raising the Hamiltonian, the helix-like complex scale factor of the branched gravitation as well as the corresponding conjugate momentum to the category of quantum operators. The resulting formulation describes the evolution of the wave function of the Universe—associated with hyper-surfaces $\Sigma_{\mathrm{ln}}$ analytically continued to the complex plane—in the cosmic scale factor $\ln^{-1}[\beta(t)]$.

Changing variable in the form $u(t) \equiv \ln^{-1}[\beta(t)]$, with $du \equiv d\ln^{-1}[\beta(t)]$, the conjugate momentum $p_{\mathrm{u}}$ of the original branching gravitation dynamical variable $\ln^{-1}[\beta(t)]$ becomes

$$p_{\mathrm{u}} = -\frac{u(t)}{N}\frac{du(t)}{dt}. \tag{7}$$

As a result of applying these standard procedures, the following branching Hamiltonian results (for the details see [13,20,24,33,41–44])

$$\mathcal{H} = \frac{1}{2}\frac{N}{u(t)}\left[-p_u^2 + g_k u^2(t) - g_\Lambda u^4(t) - g_r - \frac{g_s}{u^2(t)}\right], \tag{8}$$

with the dimensionless running coupling constants redefined as [41,42]

$$g_k \equiv \frac{2}{3\lambda - 1}; \quad g_\Lambda \equiv \frac{\Lambda M_{PI}^{-2}}{18\pi^2(3\lambda-1)^2}; \quad g_r = 24\pi^2(3g_2 + g_3);$$

$$g_s \equiv 288\pi^4(3\lambda - 1)(9g_4 + 3g_5 + g_6). \tag{9}$$

In these expression, $g_k$, $g_\Lambda$, $g_r$, and $g_s$ represent, respectively, the curvature, cosmological constant, radiation, and stiff matter coupling constant contributions. The $g_r$, and $g_s$ coupling constants can be positive or negative, without affecting the stability of the solutions. Stiff matter contribution in turn is determined by the $\rho = p$ condition in the corresponding equation of state.

The quantisation of the Lagrangian density is achieved by raising the Hamiltonian, the new dynamical variable $u(t)$ and the corresponding conjugate momentum $p_{\mathrm{u}}$ to the category of operators, represented, respectively, as $\hat{H}(t)$, $\hat{u}(t)$, and $\hat{p}_{\mathrm{u}}$:

$$\mathcal{H}(t) \to \hat{\mathcal{H}}(t); \quad u(t) \to \hat{u}(t); \quad \text{and} \quad p_{\mathrm{u}} \to \hat{p}_{\mathrm{u}} = -i\hbar\frac{\partial}{\partial u(t)}. \tag{10}$$

In what follows, for simplicity, the *hat* symbol is not used in the operators $\hat{u}$ and $\hat{p}_{\mathrm{u}}$ most of the time, as well as in most part of equations the time-dependence on the new variable $u(t)$.

Ambiguities in the ordering of the quantum operators are overcome with the introduction of a set of ordering factors, given by $\alpha = [0, 1, 2]$, following options found in the literature [32,33], as previously mentioned, with $p^2$ defined as

$$p^2 \equiv -\frac{1}{u^\alpha(t)}\frac{\partial}{\partial u(t)}\left(u^\alpha(t)\frac{\partial}{\partial u(t)}\right). \tag{11}$$

The approach based on the insertion of a set of ordering factors, makes it possible to obtain a broader class of solutions for the Universe wave function.

Combining (8) and (11), we get the subsequent expression for the Wheeler–DeWitt equation for the wave function of the Universe, $\Psi(t)$:

$$\mathcal{H}(t)\Psi(u) = \left(-\frac{1}{u^\alpha}\frac{d}{du}\left(u^\alpha\frac{d}{du}\right) + V(u)\right)\Psi(u) = 0 \tag{12}$$

with the effective potential[10]

$$V(u) = -\eta_r + \eta_m u + \eta_k u^2 + \eta_q u^3 - \eta_\Lambda u^4 - \frac{\eta_s}{u^2}, \tag{13}$$

which we supplemented with two additional terms, $\eta_m u$, that describes the contribution of baryon matter combined with dark matter, and $\eta_q u^3$, a quintessence-term. From this expression, for $\alpha = 0$, we obtain the following equation under the action of a real potential[11] represented by $V(u)$:

$$\left( -\frac{d^2}{du^2} + V(u) \right) \Psi(u) = 0. \tag{14}$$

With the choice $\alpha = 1$ in expression (12), we get the equation

$$\left( -\left\{ \frac{1}{u}\frac{d}{du} + \frac{d^2}{du^2} \right\} + V(u) \right) \Psi(u) = 0. \tag{15}$$

Finally, the choice $\alpha = 2$ in expression (12), results in the following equation

$$\left( -\left\{ \frac{2}{u}\frac{d}{du} + \frac{d^2}{du^2} \right\} + V(u) \right) \Psi(u) = 0. \tag{16}$$

With a view to comparing results based on the standard formulation, in what follows, we set up the dimensionless coupling parameters of the effective potential with values found in the literature, complementing the coupling constants of baryon and dark matter and quintessence with a parametrization based on the total density parameter, $\Omega_0$, which describes the ratio between the total average density of matter and energy in the early Universe, $\rho_T$ and the critical density, $\rho_{crit}$. The most accepted value of the density parameter nowadays is:

$$\Omega_0 \equiv \frac{\rho_T}{\rho_{crit}} = \quad \Omega_B + \Omega_{DM} + \Omega_\Lambda \sim 0.04 + 0.23 + 0.73 \sim 1, \tag{17}$$

where $\Omega_B$, $\Omega_{DM}$, and $\Omega_\Lambda$ represent the baryon matter, dark matter and dark energy density parameters, respectively. At this stage of our investigation, we do not intend to obtain numerical data that may support future cosmological observations, but rather to seek first to establish a formal consistency in the treatment of the quantum branch-cut gravitation, with the aim of establishing observational predictions based on a consistent theoretical formulation in the future. There are numerous formulations in the literature, based on standard cosmology, that consistently deal with this problem, using improved technical models. Just to name a few of these, we indicate [13,20,24,27,33,41–44], among many others.

Figures 2–5 show the behavior of the effective potential for sets of values of the running coupling constants. The behavior of the effective potential shows a domain of the stiff matter term, contributing for the presence of singularities, both in the expansion and contraction regions. These results show that the increase of the running coupling constants of the stiff matter produces an enlargement of the singularity domain region. Evidently, a more rigorous analysis of the role and consistency of these parameterizations is necessary. For example, the adoption of coupling constants based on energy density parameter [33], or, in the absence of the stiff matter, to examine the relative contributions of the other contributions. In any case, a lesson learned from this work is the need to seek formal alternatives for the inclusion of such contributions so as not to reinforce, —although such a conclusion is far from categorical—, in an artificial and inconsistent way the dominance of certain alternatives over others.

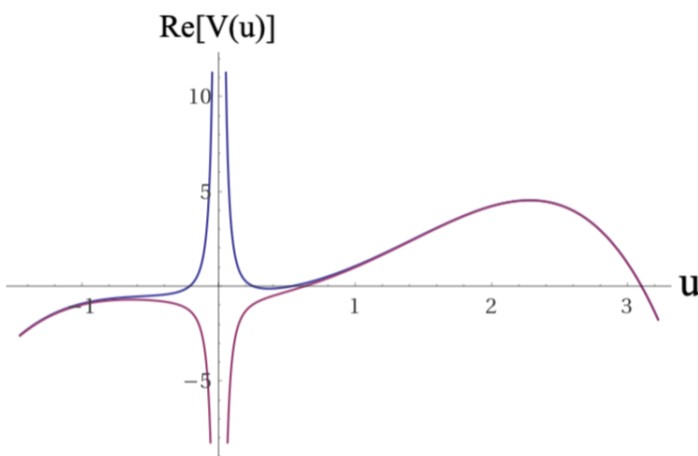

**Figure 2.** Plot of the real part of the potential defined in Equation (13). In the top figure the coupling constants values are: $\eta_r = 0.6$, $\eta_m = 0.2855$, $\eta_k = 1$, $\eta_q = 0.7$, $\eta_\Lambda = 1/3$, and $\eta_s = -0.03$. In the bottom figure the coupling constants values are: $\eta_r = 0.6$, $\eta_m = 0.2855$, $\eta_k = 1$, $\eta_q = 0.7$, $\eta_\Lambda = 1/3$, and $\eta_s = +0.03$. Values of parameters taken from [43,46,47].

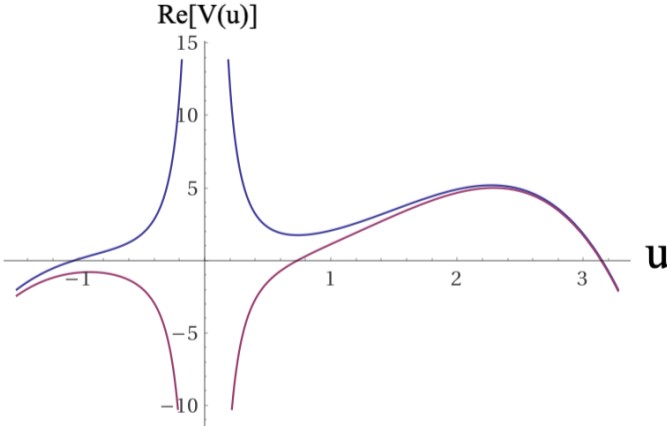

**Figure 3.** Similar plot of the previous figure. Coupling constants values in the top figure: $\eta_r = 0.024$, $\eta_m = 0.2855$, $\eta_k = 1$, $\eta_q = 0.7$, $\eta_\Lambda = 1/3$, and $\eta_s = -0.468$. Coupling constants values in the bottom figure: $\eta_r = 0.024$, $\eta_m = 0.2855$, $\eta_k = 1$, $\eta_q = 0.7$, $\eta_\Lambda = 1/3$, and $\eta_s = +0.468$. Values of parameters taken from [43,46,47].

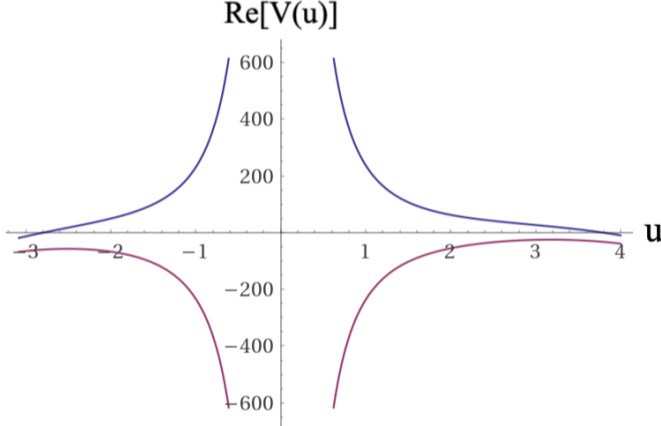

**Figure 4.** Similar plot of the previous figure. Coupling constants values in the top figure: $\eta_r = 0.0$, $\eta_m = 0.2855$, $\eta_k = 1$, $\eta_q = 0.7$, $\eta_\Lambda = 1/3$, and $\eta_s = -234.0$. Coupling constants values in the bottom figure: $\eta_r = 0.0$, $\eta_m = 0.2855$, $\eta_k = 1$, $\eta_q = 0.7$, $\eta_\Lambda = 1/3$, and $\eta_s = +234.0$. Values of parameters taken from [43,46,47].

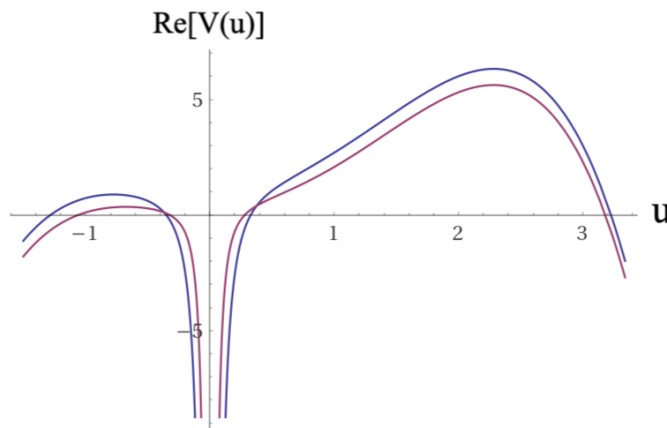

**Figure 5.** Similar plot of the previous figure. Coupling constants values in the top figure: $\eta_r = -1.22$, $\eta_m = 0.2855$, $\eta_k = 1$, $\eta_q = 0.7$, $\eta_\Lambda = 1/3$, and $\eta_s = 0.15$. Coupling constants values in the bottom figure: $\eta_r = -0.5$, $\eta_m = 0.2855$, $\eta_k = 1$, $\eta_q = 0.7$, $\eta_\Lambda = 1/3$, and $\eta_s = 0.05$. Values of parameters taken from [43,46,47].

### 3.1. Complex Conjugation of the Friedmann's-Type Wave Equations

In the branching gravitation, the Friedmann's-type equations, analytically continued to the complex plane, and expressed in terms of the new variables $u(t)$, are [7–9]:

$$\left(\frac{\frac{d}{dt}u(t)}{u(t)}\right)^2 = \frac{8\pi G}{3}\rho(t) - \frac{kc^2}{u(t)} + \frac{1}{3}\Lambda, \tag{18}$$

and

$$\left(\frac{\frac{d^2}{dt^2}u(t)}{u(t)}\right) = -\frac{4\pi G}{3}\left(\rho(t) + \frac{3}{c^2}p(t)\right) + \frac{1}{3}\Lambda, \tag{19}$$

where $\Lambda$ represents the cosmological constant (see Section 1). The corresponding complex conjugated Friedmann's-type equations are:

$$\left(\frac{\frac{d}{dt}u^*(t^*)}{u^*(t^*)}\right)^2 = \frac{8\pi G}{3}\rho^*(t^*) - \frac{kc^2}{u^*(t^*)} + \frac{1}{3}\Lambda^*, \tag{20}$$

and

$$\left(\frac{\frac{d^2}{dt^2}u^*(t^*)}{u^*(t^*)}\right) = -\frac{4\pi G}{3}\left(\rho^*(t^*) + \frac{3}{c^2}p^*(t^*)\right) + \frac{1}{3}\Lambda^*. \tag{21}$$

Equations (18)–(21) underlie the scenarios of branched gravitation in the imaginary sector, as discussed before (see Figure 6): in the first scenario, in the region before the primordial singularity, there is a continuous evolution of the Universe around a branch-cut in the transition region as a function of an imaginary time parameter, conjugated to the corresponding time parameter of the later evolutionary region and no primordial singularity occurs; in the second scenario, the branch-cut and the branch point disappear after realization of the imaginary time by means of a Wick rotation, then this parameter is replaced by the real and continuous thermal time, the temperature. As a result, a parallel evolutionary mirror universe, adjacent to our own, is nested in the fabric of space and time, with its evolutionary process receding into the cosmological sector of negative thermal time. In the following, we adopt, as a consistent formal procedure, conjugated complex versions of expressions (14)–(16). Furthermore, as a consequence of this procedure, solutions of the wave function of the Universe that describe the quantum evolution (in the cosmic scale parameter $\ln^{-1}[\beta(t)]$) of the scenarios described above can be obtained.

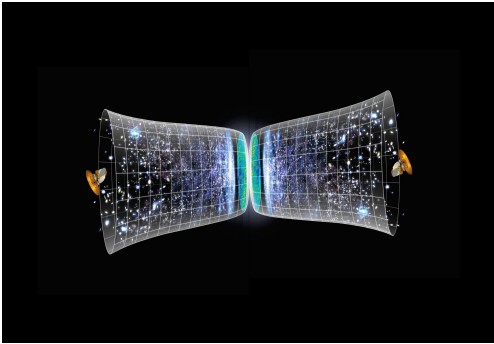
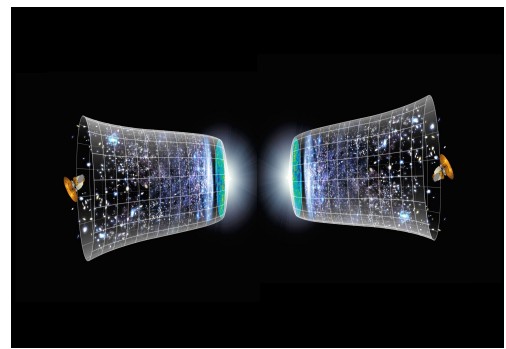

**Figure 6.** Artistic representations of the cosmic contraction and expansion phases of the branch-cut universe evolution scenarios. On the left figure, the branch-cut universe evolves from negative to positive values of the imaginary cosmological time $t_i$, circumventing continuously the branch-cut and no primordial singularity occurs, only branch points. On the right figure the branch-cut and branch point disappear after the realisation of imaginary time by means of a Wick rotation, which is replaced here by the real and continuous thermal time (temperature), $T$. In this scenario, a mirrored parallel evolutionary universe, adjacent to ours, is nested in the structure of space and time, with its evolutionary process going backwards in the cosmological thermal time negative sector. Figures based on artistic impressions [48].

### 3.2. Solutions and Boundary Conditions

The boundary conditions adopted in this work follows the conventional canons of convergence, as well as stability and continuity of the solutions of the differential equations. Moreover, as a new topic in this contribution, we analyze the boundary conditions of the wave function of the Universe in the light of the Bekenstein criterion [36].

The impossibility of packing the energy and entropy of the primordial Universe into finite dimensions considering spatially connected regions within the particle horizon of a given observer, locus of the most distant points that can be observed at a specific time $t_0$ in an event, made Bekenstein [36] conjecture an upper bound, given by $\frac{2\pi R}{\hbar c}$, for the entropy $S$ and energy $E$ of a system contained in a spherical region of radius $R$:

$$\frac{2\pi R}{\hbar c} \geq S/E \quad \text{so} \quad S \leq S_B = \frac{2\pi}{\hbar c} ER, \tag{22}$$

in which $S_B$ denotes the upper limit of Bekenstein entropy.

Considering in a simplified way the proper distance $d(t)$ of a pair of objects, in an arbitrary time $t$ and its relationship with the proper distance $d(t_0)$ in a reference time $t_0$, $d(t) = u(t)d(t_0)$, this implies that for $t = t_0$, $u(t_0) = 1$. We consider the boundary condition $|u(t_0) = 1|$, assuming the time $t_0$ as the locus of the most distant points that can be observed, in tune with the Bekenstein criterion. With this assumption, due to the structural characteristics of the proposed effective potential and the extended class of solutions for the wave equations, the wave function of the Universe obeys the following boundary conditions in the expansion sector of the primordial Universe: $\Psi(1) = 1, \Psi'(1) = 0$ and $\Psi(1) = 0, \Psi'(1) = 1$. Similarly, in the contraction sector of the primordial Universe, we have the boundary conditions: $\Psi(-1) = -1, \Psi'(-1) = 0$ and $\Psi(-1) = 0, \Psi'(-1) = -1$, in opposition to the "no boundary" condition [25].

In Figure 7, we plot a sampling solutions family of Equation (14) corresponding to the expansion region of the universe, using a set of values from [43,46,47]. The solutions are in agreement with the corresponding results presented in the literature, although we have not resorted, unlike other authors, to approximations to solve the corresponding differential equations. Approaches adopted by other authors, based on approximations, mainly in the primordial singularity region, limit their numerical analysis, although they have not significantly influenced the global and oscillatory behavior of the solutions.

In Figures 8–13 we show the solutions of Equations (14)–(16). As shown in the figures, for the region domains between $u = -1$ and $u = 1$ the differential equations

have no solutions. In our interpretation, this domain corresponds to the region in which a topological quantum leap occurs in accordance with the Bekenstein criterion [11,13].

The main characteristics of these solutions are the oscillatory behavior, whose amplitudes are decreasing as the universe expands, implying an Universe described by oscillating quantum states tending toward a stable ordering at some future time. In the opposite direction, the systematic increase of the oscillatory amplitudes of the wave function as a function of the scale factor $\ln^{-1}[\beta(t)]$ suggests the accumulation of branches, as indicated by the BCGT for restoration of causality. The effect of accumulating branches actually occurs in both the expansion and contraction regions near the transition region.

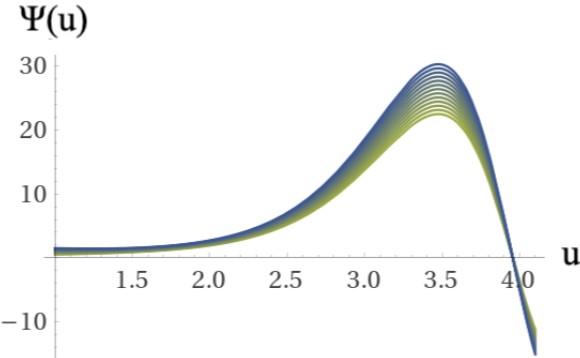

**Figure 7.** Sampling solution family of Equation (14) with the values of the coupling constants: $\eta_r = 0.6$, $\eta_m = 0.2855$; $\eta_k = 1$; $\eta_q = 0.7$; $\eta_\Lambda = 1/3$; $\eta_s = -0.03$. Values of parameters taken from [43,46,47].

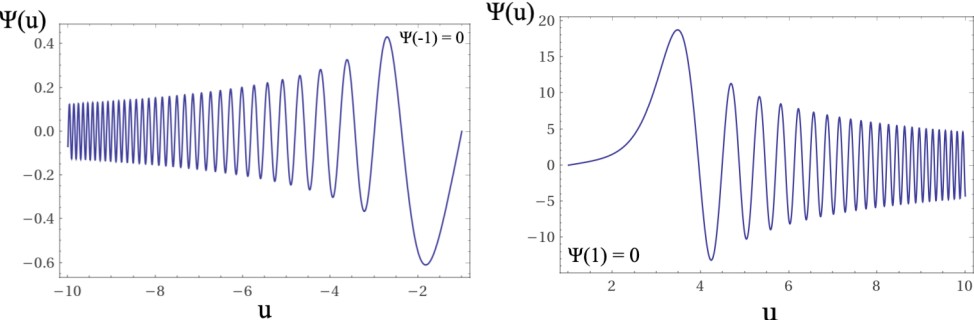

**Figure 8.** Solutions of Equation (14). The values of the coupling constants are: $\eta_r = 0.6$, $\eta_m = 0.2855$, $\eta_k = 1$, $\eta_q = 0.7$, $\eta_\Lambda = 1/3$, and $\eta_s = -0.03$. Values of parameters taken from [43,46,47].

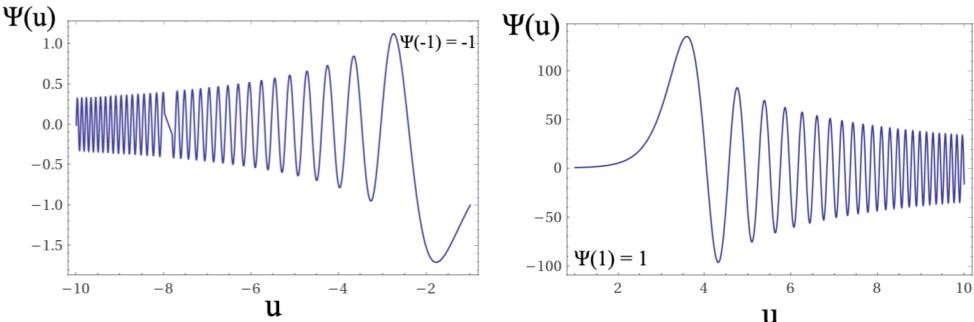

**Figure 9.** Solutions of Equation (14). The values of the coupling constants are: $\eta_r = -1.22$, $\eta_m = 0.2855$, $\eta_k = 1$, $\eta_q = 0.7$, $\eta_\Lambda = 1/3$, and $\eta_s = 0.15$. Values of parameters taken from [43,46,47].

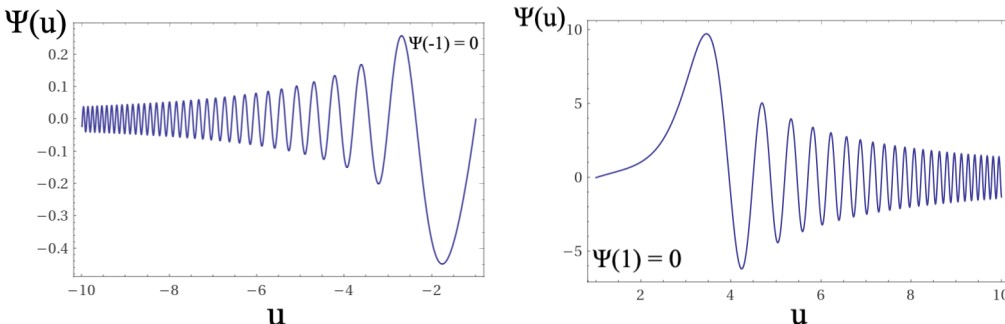

**Figure 10.** Solutions of Equation (15). The values of the coupling constants are: $\eta_r = 0.6$, $\eta_m = 0.2855$, $\eta_k = 1$, $\eta_q = 0.7$, $\eta_\Lambda = 1/3$, and $\eta_s = -0.03$. Values of parameters taken from [43,46,47].

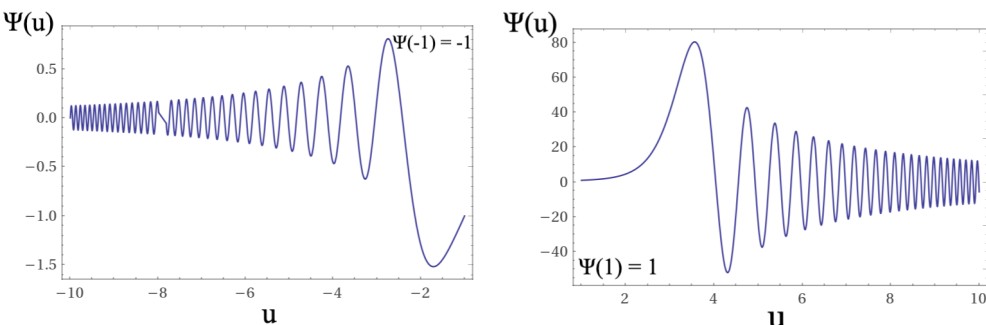

**Figure 11.** Solutions of Equation (15). The values of the coupling constants are: $\eta_r = 0.6$, $\eta_m = 0.2855$, $\eta_k = 1$, $\eta_q = 0.7$, $\eta_\Lambda = 1/3$, and $\eta_s = -0.03$. Values of parameters taken from [43,46,47].

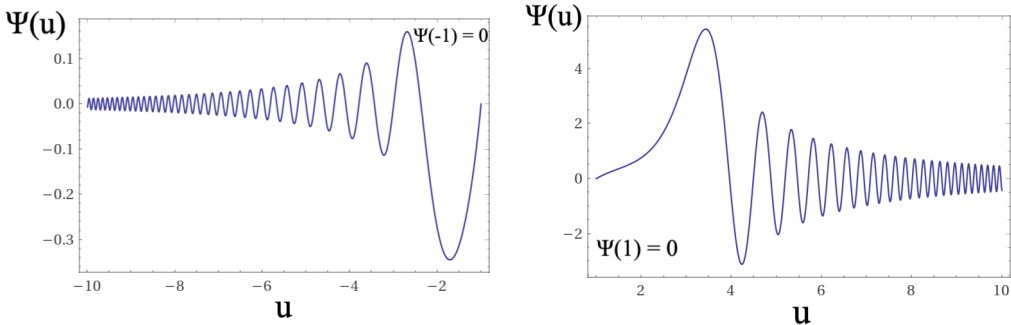

**Figure 12.** Solutions of Equation (16). The values of the coupling constants are: $\eta_r = 0.6$, $\eta_m = 0.2855$, $\eta_k = 1$, $\eta_q = 0.7$, $\eta_\Lambda = 1/3$, and $\eta_s = -0.03$. Values of parameters taken from [43,46,47].

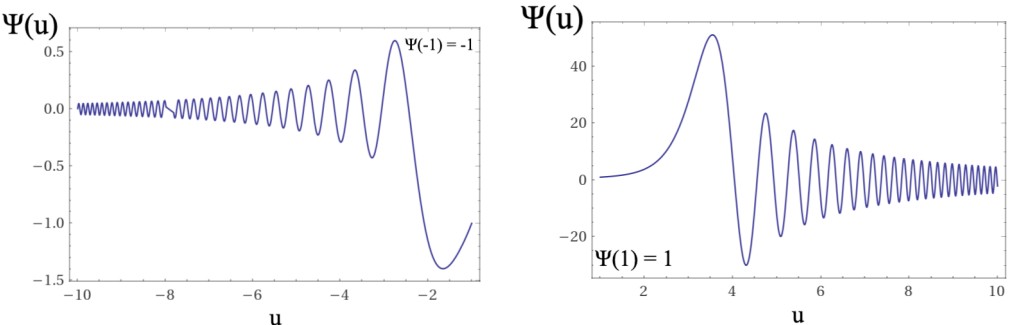

**Figure 13.** Solutions of Equation (16). The values of the coupling constants are: $\eta_r = 0.6$, $\eta_m = 0.2855$, $\eta_k = 1$, $\eta_q = 0.7$, $\eta_\Lambda = 1/3$, and $\eta_s = -0.03$. Values of parameters taken from [43,46,47].

### 4. Conclusions

We summarize our most relevant results. We adopt as an underlying proposition a compact universe, filled with homogeneous matter, which exists forever in a quantum state, either static or oscillating, without imposing in an ad hoc way a restriction limit for the cosmological scale factor and for the wave function of the Universe, as its disappearance at any limit of the scale factor (see Ref. [49]). Although its disappearance occurs naturally in the transition region of the branched model, as a natural consequence of the imposition of the Bekenstein criterion, in the expansion and contraction phases, the oscillatory behavior of the wave function of the Universe is characterized by an increase in its amplitudes in the anterior region of the transition phase, indicating consistency with the proposition of accumulating branches to reestablish causality. In the expansion region, going back in time, the same effect occurs. These results indicate that in the limit $u(t) \to \infty$ (or $\ln^{-1}[\beta(t)] \to \infty$), $\Psi(u) \to 0$, implying a Universe described by oscillating quantum states tending towards a stable configuration at some future time. The opposite behavior is verified in the mirror sector of the model. In the mirror sector, the Universe evolves from a stable to an unstable quantum state, and in the visible sector, from an unstable to a stable quantum state. This behavior is contrary to the entropy behavior of the system, which decreases in the evolutionary process of the mirror universe and increases in the visible sector. Making these phenomena compatible seems a challenging task.

Our interpretation of the disappearance of the wave function of the Universe, in turn, in the region between $u = -1$ and $u = 1$, where a topological quantum leap or tunneling occurs according to Bekenstein's criterion, although with a certain harmony with the Vilenkin's quantum tunneling proposal [50], differs from most known proposals for the corresponding boundary conditions[12]. This is because these proposals, although based on different conceptions and assumptions, have in common the prediction of an inflationary stage of evolution in order to reconcile the causality problem of the primordial Universe. In turn, causality involving the horizon size and the patch size, as stressed before, may be accomplished in branch-cut cosmology through the accumulation of branches in the transition region between the present state of the Universe and the past events [10].

The hypothesis of the isotropy of the branch-cut Universe, one of the pillars of cosmology, and its mirror partner may be questioned based on deviations observed in recent decades by means of cosmological probes [52]. Anisotropy of the Universe, in our conception, has two branches to be approached. One branch refers to the evolutionary anisotropy of the mirror universe to our own. Furthermore, another, to the anisotropic directional evolution in both universes. This is a topic that deserves systematic study in the future. Although it is still early for a more effective direction in this study, some aspects deserve attention, such as, for example, the consequences of adopting a non-symmetric approach and a different ordering of the dimensionless thermodynamics connection $\epsilon$, the role of dark matter in the evolution of the branch-cut universe, the role of fluctuations in the primordial spectrum and seeds in the the early universe, and also questions regarding the multiverse content. Likewise, alternative models that address this issue in a complementary way to ours, such as the bouncing model of Ijjas, Steinhardt, and Loeb [17], or Belinsky and Khalatnikov [42,53] proposition for a generic solution of the Einstein equations near their cosmological singularity, based on a generalization of the homogeneous model of Bianchi type IX, deserve our attention in the near future.

The presented proposal strengthens the idea of the transition region of the branched Universe acting as a 'portal' for cosmic material, playing the role this way of an 'eternal seed' [54] for the expanding emergent cosmic scenario.

Finally, a peculiar aspect of the class of solutions presented concerns the insertion of the operators ordering parameter $\alpha$. As we can see in the presented solutions (Figures 8–13), different values of $\alpha$, in combination with different choices of running coupling constants affect the amplitudes of the wave function of the Universe and therefore, according to our interpretation, the accumulation of branches in order to restore causality. Evidently,

the results presented are still at a preliminary stage of investigation, requiring a more systematic approach in order to broaden its scope.

The conclusions of this work lead to numerous underlying questions, whose understanding has motivated in-progress investigations.

**Author Contributions:** Conceptualization, C.A.Z.V.; methodology, C.A.Z.V. and B.A.L.B. and P.O.H.B. and J.A.d.F.P. and D.H.; software, C.A.Z.V. and B.A.L.B. and M.R.; validation, C.A.Z.V. and B.A.L.B. and D.H. and P.O.H.B. and J.A.d.F.P.; formal analysis, C.A.Z.V. and B.A.L.B. and P.O.H.B. and J.A.d.F.P. and D.H.; investigation, C.A.Z.V. and B.A.L.B. and P.O.H.B. and J.A.d.F.P. and M.R. and G.A.D.; resources, C.A.Z.V.; data curation, C.A.Z.V. and B.A.L.B.; writing—original draft preparation, C.A.Z.V.; writing—review and editing, C.A.Z.V. and B.A.L.B. and P.O.H.B. and J.A.d.F.P. and D.H. and G.A.D. and M.R.; visualization, C.A.Z.V. and B.A.L.B.; supervision, C.A.Z.V.; project administration, C.A.Z.V.; funding acquisition (no funding acquisition). All authors have read and agreed to the published version of the manuscript.

**Funding:** This research received no external funding.

**Institutional Review Board Statement:** Not applicable.

**Informed Consent Statement:** Not applicable.

**Data Availability Statement:** Not applicable.

**Acknowledgments:** P.O.H. acknowledges financial support from PAPIIT-DGAPA (IN100421). The authors wish to thank the referees for suggestions that enabled substantial improvement of this article.

**Conflicts of Interest:** The authors declare no conflict of interest.

## Notes

[1] Hawking and Hertog, in 2018, revisited the multiverse concept, conjecturing that the output of eternal inflation does not produce an infinite fractal-type multiverse, but is finite and reasonably smooth.

[2] For simplicity the cosmological constant term has been suppressed.

[3] We emphasize that these equations do not represent a direct parameterization or generalization of the conventional Friedmann equations described in a single-pole metric and likewise the new cosmic scale factor does not represent a simple parameterization of the standard theory scale factor. Due to the non-linearity of Einstein's equations, such a direct generalization or parametrization would be inconsistent. For the details, see [7–9,12].

[4] The impossibility of packing energy and entropy according to the Bekenstein Criterion into a finite size makes the transition phase between contraction and expansion very peculiar, imposing a topology where space-time shapes itself topologically around a branch point.

[5] The Hořava–Lifshitz (HL) formulation main goal is to get a renormalizable theory by means of higher spatial-derivative terms of the curvature which are added to the Einstein–Hilbert action [20]. A recurring problem addressed in the analysis of the Hořava–Lifshitz theory of gravity is related to the preservation of general diffeomorphism, a fundamental constraint of general relativity [22]. Although this is not the main topic of discussion, we would like to address that, in the case of restricted foliation preserving diffeomorphism invariance of the Hořava–Lifshitz theory, a well behaved Hamiltonian for gravity may be found [23].

[6] For an interesting discussion of this topic see Ref. [31].

[7] We emphasize once more that $\ln^{-1}[\beta(t)]$ represents the reciprocal of $\ln[\beta(t)]$ and $\beta(t)$ identifies the range and cuts of the helix-like cosmological factor in branched gravitation. $\ln^{-1}[\beta(t)]$ characterizes complex topological leafs of singular foliations by means of Riemann surfaces.

[8] $N(t)$ does note represent a dynamical quantity; in turn it denotes a pure gauge variable.

[9] As is well know, there are several quantization methods, as for instance, the canonical quantization and the related Dirac scheme, Segal and Borel quantizations, geometric quantization, various ramifications of deformation quantization, Berezin and Berezin–Toeplitz quantizations, prime quantization and coherent state quantization. For a broad overview see [45]. The advantage of the canonical procedure to quantize a classical theory resides in the preservation of the original formal structure, symmetries and conservation laws. The denomination 'spacetime topological canonical quantization' is due to the combination of the conventional canonical quantization procedure applied to a variable, the helix-like complex cosmic scale factor of the branched gravitation, $u = \ln^{-1}[\beta(t)]$, raised to the category of quantum operator, which presents an intricate topology.

[10] The condition $\mathcal{H}\Psi(t) = 0$ excludes the multiplicative term $\frac{1}{2}\frac{N}{u(t)}$ in Equation (8).

[11]    Despite that we consider only the real part of the effective potential, the variable $u$ is complex, and the solutions still have a broader scope, describing the behavior of the wave function of the Universe both for the contraction region, prior to the primordial singularity, and for the later expansion cosmological region.

[12]    The tunneling boundary condition of Vilenkin [51] in particular has two degrees of freedom: the scale factor and a homogeneous scalar field. A tunneling wave function then describes an ensemble of universes tunneling from "nothing" to a de Sitter space, and then evolving along the lines of an inflationary scenario and eventually collapsing to a singularity [51].

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
