# Peer review of "A Wheeler–DeWitt Quantum Approach to the Branch-Cut Gravitation with Ordering Parameters"

_universe, doi:10.3390/universe9060278_

Round 1

Reviewer 1 Report

In this work a formulation of quantum gravity based on the Wheeler-DeWitt (WDW) equation combined with the classical concepts of the branch-cut cosmology, which contemplates as a new scenario for the origin of the Universe, a smooth transition region between the contraction phase, prior to the primordial singularity, and the subsequent expansion phase are investigated. It should be noted that the consideration is not carried out within the framework of the usual theory of gravity based on the Hilbert-Einstein action, but in the framework of a modified Horava-Lifshitz  theory of gravity with the action, given by SHL, that employs terms dependent on the scalar curvature of the Universe and its derivatives, in different orders with many additional parameters.

Our remarks.

1.         The authors additionally introduce into the theory the contributions of baryonic matter, dark matter, and quintessence with the corresponding arbitrary parameters, however, without any discussion of them.

2.         The authors consider the ambiguity in the problem of ordering noncommuting factors as an extension of the class of feasible solutions for the wave function of the universe. At the same time, the limited choice of exponents  is unclear.

3.    The common factor  in the Hamiltonian (6) is excluded from further consideration without any justification, although it could be taken into account in the definition of kinetic energy, as well as in the problem of ordering.

4.     What can the authors say about a possibility of taking into account the anisotropy of the universe in their approach (the Bianchi IX model)?

We recommend the paper for publication after minor revision.

Author Response

Message to the referees: We thank the referees for their attention and careful revision of the text. Your comments were very useful for improving the text, for which we are very grateful. As for the observations, we indicate the location of our modifications for better monitoring of them within the following text, in blue. Initially, we inform you that, in order to meet the observations, the text has been expanded, with the insertion of a new introductory section and the shift from the previous introductory section to the following section. Furthermore, some references have been incorporated into the original text.

Reviewer 2 Report

The refereed paper is devoted to an interesting and important topic of the treatment of the Wheeler-DeWitt equation. They declare that they have invented a new approach to cosmology and gravity, which resolves many problems, but it is difficult to understand reading the paper, because it is not self-consistent and is not clearly written. They introduce different notions without explanation. It is not clear why they can use the complex coordinates. In some places they use the term "multiverse" without any comment. They use Horava - Lifshitz gravity which is essentially different from the genarla relativity and do not even mention this fact in the abstract and in the introduction. Besides, it is not clear if the geometrodynamic approach and the Wheeler-DeWitt equation are applicable in the framework of the Horava - Lifshitz gravity, where the general diffeomorphism invariance is absent. Besides, the authors do not try to compare their approach with other present in the literature (for an extensive review, see e.g. the monography "Quantum Gravity" by C. Kiefer). I think that the paper cannot be published in its present form and should be essentially improved. 

Author Response

(The authors gave the same response as above.)

Reviewer 3 Report

This manuscript is an extended analysis of the authors’ initial published research on the topic of quantization of a branch-cut cosmology with a Wheeler-DeWitt approach. This time, they considered ordering factors in the quantization of the conjugated momentum and expanded their initial report by adding the contribution of baryon matter, dark matter, and quintessence fields, considering Bekenstein criterion for the entropy-to energy ration to constrain their boundary conditions. This is a valid improvement over their previous work, but a few issues must be addressed for this paper to work on its own.

1)      In the introduction, the authors mentioned they have proposed a topological canonical quantum approach for the classical branch-cut cosmology (line 32). I suggest they summarize their previous work, explaining this approach and how it compares to other quantization methods (extending their literature review in the process).

2)      The authors said the spacetime topological quantisation results in a formulation that describes the evolution of the wave-function (line 88) even though the Wheeler-DeWitt equation (10) does not depend on a time parameter.  The WdW equation is not a Schrödinger-type equation (as the authors claim in line 117) since the time evolution is missing. This is the well-known problem of time (see Isham https://arxiv.org/abs/gr-qc/9210011). Therefore, a deeper discussion on what evolution of the wave-function means is needed. Is the wave-function evolving with respect to what?

3)      With respect to the ordering factors (line 108), why was alpha restricted to the subset [0,1,2]? It is not true that non-integer values for alpha have no significance. In fact, alpha can assume any real value (see, for example, https://arxiv.org/abs/gr-qc/0511149v1).

4)      A discussed interpretation of the plots presented in Figures 1-4 is needed.

5)      Why is the word naturally emphasised (twice) in Subsection 3.4 and Section 4?

6)      Reference 13 has the wrong year of publication.

The writing is adequate, although a few stylistic choices should be reconsidered. The first is the repeated appearance of the word "novelty" throughout the text, which can be misleading. The results are an extension of the author's previous work, thus the novelty refers to a comparison with these papers instead of the whole literature. Also, using italics and single quotes to emphasise the word naturally in subsection 3.4 and section 4 is overkill.

Author Response

(The authors gave the same response as above.)

Round 2

Reviewer 2 Report

The authors have taken into account my suggestions and I think that the paper can be published in its present form.

Reviewer 3 Report

I have revised this version of the manuscript and the author's reply to my comments. The author's modifications adequately answer the issues detected in the previous version. (The only thing I do not agree with is being considered to be a man. :) I recommend the paper for publication.